# Simplifying the development of portable, scalable, and reproducible workflows

**Stephen R Piccolo[1]\*, Zachary E Ence[1], Elizabeth C Anderson[1], Jeffrey T Chang[2], Andrea H Bild[3]**

[1]Department of Biology, Brigham Young University, Provo, United States; [2]Department of Integrative Biology and Pharmacology, University of Texas Health Science Center at Houston, Houston, United States; [3]Department of Medical Oncology and Therapeutics, City of Hope Comprehensive Cancer Institute, Monrovia, United States

**Abstract** Command-line software plays a critical role in biology research. However, processes for installing and executing software differ widely. The Common Workflow Language (CWL) is a community standard that addresses this problem. Using CWL, tool developers can formally describe a tool's inputs, outputs, and other execution details. CWL documents can include instructions for executing tools inside software containers. Accordingly, CWL tools are portable—they can be executed on diverse computers—including personal workstations, high-performance clusters, or the cloud. CWL also supports workflows, which describe dependencies among tools and using outputs from one tool as inputs to others. To date, CWL has been used primarily for batch processing of large datasets, especially in genomics. But it can also be used for analytical steps of a study. This article explains key concepts about CWL and software containers and provides examples for using CWL in biology research. CWL documents are text-based, so they can be created manually, without computer programming. However, ensuring that these documents conform to the CWL specification may prevent some users from adopting it. To address this gap, we created ToolJig, a Web application that enables researchers to create CWL documents interactively. ToolJig validates information provided by the user to ensure it is complete and valid. After creating a CWL tool or workflow, the user can create 'input-object' files, which store values for a particular invocation of a tool or workflow. In addition, ToolJig provides examples of how to execute the tool or workflow via a workflow engine. ToolJig and our examples are available at https://github.com/srp33/ToolJig.

**\*For correspondence:**
stephen_piccolo@byu.edu

**Competing interest:** The authors declare that no competing interests exist.

## Introduction

Software is fundamental to modern scientific research (**Hey et al., 2009**; **Wilson, 2014**). It can accelerate the pace of research, formalize algorithmic logic, and support reproducibility (**Piccolo and Frampton, 2016**). In a 2014 survey, 92 % of academic scientists reported using software in their research (**Hong, 2014**). Our article focuses on command-line *tools*, which scientists use in many disciplines (**Kumar and Dudley, 2007**) and which provide advantages over point-and-click tools. In particular, the process of executing command-line tools can be formalized (**Kumar and Dudley, 2007**; **Piccolo and Frampton, 2016**). Our article also focuses on computational *workflows*, which are defined series of steps for processing or analyzing data (**Leipzig, 2017**). Each step in a workflow applies one or more command-line tools to the data with specific inputs, outputs, and configuration settings. A *workflow engine* is a software system for executing workflows. For example, the Snakemake and Nextflow workflow engines facilitate execution of workflows and are used widely for scientific research (**Di Tommaso et al., 2017**; **Köster and Rahmann, 2012**). These and other workflow engines provide flexibility regarding the computing environment in which a workflow is executed, allowing researchers

to use local, cluster-, or cloud-based computers. In many cases, workflow steps can be executed in parallel. With this flexibility comes the challenge of ensuring that operating system and tool configurations are consistent across all computers used. This process is made easier when workflows provide instructions for executing the steps within software containers (*Boettiger, 2015*).

In scientific research, the use of workflows can be classified into two main categories. One category includes *orchestration* systems, in which workflow engines repeatedly process data of a given type. For example, a genomics core facility might use a workflow to ingest DNA-sequencing data, align reads to a reference genome, and call DNA variants, as well as other steps in between. The core facility might have other workflows for processing RNA-sequencing or bisulfite-sequencing data. In some cases, a modular design might be useful; for example, some command-line tools could be reused in multiple workflows. In this scenario, data throughput is a key concern, so the workflow engine might be connected to cluster- or cloud-based computer environments, allowing throughput to scale as needs fluctuate. Arvados and Tibanna are examples of cloud-based orchestration systems (*Amstutz, 2015*; *Lee et al., 2019*). Scientists might also take advantage of error handling provided by some workflow engines. For example, if a power outage occurred while a workflow was being executed, it would be possible to restart the workflow at the point of failure rather than needing to rerun the entire workflow. A second way that workflows are used in scientific research is to support reproducibility of a particular analysis. When performing research studies, scientists often use existing software and write custom code to process data, apply statistical tests, create graphics, etc. Inherently, the particular combination of computational tasks used in a given study should be unique. Therefore, there is no need to orchestrate the tasks for repeated execution. However, much benefit can come from sharing the code with the scientific community and ensuring that others can easily re-execute the code (*Piccolo and Frampton, 2016*). Workflows aid in this process because they provide a way to encapsulate the logic for executing code in a fully specified computational environment with all necessary software dependencies installed. For many studies, a relatively simple workflow engine like *cwltool* (RRID:SCR_015528) can be used to re-execute the analyses.

Different workflow engines use different methodologies and vocabularies for defining workflows and for interfacing with software within those workflows (*Leipzig, 2017*). For example, to use Snakemake or Nextflow, a researcher can specify a workflow using a custom programming language that is specific to either workflow engine. These languages are extensions of Python and Groovy, respectively. Python and Groovy are familiar to many researchers, making them relatively easy for these researchers to adopt. However, if a researcher does not have programming expertise, is not familiar with these programming languages, or wants to switch between workflow engines, workflow creation might be difficult or time-consuming. Such challenges motivated creation of the Common Workflow Language (CWL), a formal specification for describing command-line tools and workflows (*Amstutz et al., 2016*). The CWL project is a community-based, collaborative effort driven by individuals and institutions across diverse disciplines; participation is open to anyone. These efforts should help to ensure the project's longevity and acceptance among researchers. Indeed, CWL documents are already recognized by many workflow engines, including Snakemake (*Köster and Rahmann, 2012*), *cwltool*, Toil (*Vivian et al., 2017*), Apache Airflow (*Kotliar et al., 2019*), Tibanna (*Lee et al., 2019*), and Arvados (*Amstutz, 2015*).

By creating CWL documents, scientists can describe tools and workflows in a way that is standards-based and agnostic to the workflow engine(s) on which they are executed. CWL documents are text-based files and thus can be created via a text editor without doing any computer programming. Despite this simplicity, many researchers have yet to adopt CWL. Useful tutorials are available online (*Common Workflow Language working group, 2021*), but the CWL specification provides great flexibility, so researchers face a learning curve to ensure that documents are specified correctly. Some software applications are available to aid in this process. For example, *Rabix Composer* is a desktop application that enables researchers to create and edit CWL documents (*Amstutz et al., 2021*). This application supports many features in the CWL specification and provides text-based as well as visual editors. However, researchers unfamiliar with nuanced details of the CWL specification may find Rabix Composer too advanced for their needs, and it does not support the latest versions of CWL. Alternatively, CWL plugins exist for many code editors (*Common Workflow Language working group, 2021*). In addition, developers have created application programming interfaces that enable

researchers to build CWL documents. However, many researchers who could benefit from CWL lack the programming expertise to use these resources.

In this article, we illustrate how to use CWL to describe command-line tools and workflows and to perform reproducible research analyses. We provide 27 examples of CWL documents for completing diverse types of research tasks, ranging from simple (e.g., printing custom messages to the console) to advanced (identifying differentially expressed genes or calling somatic mutations in cancer genomes). In addition, we introduce *ToolJig*, a Web application that enables researchers to create CWL documents. *ToolJig* provides a simple, interactive interface that requires no installation and includes prompts to guide the user. Via *ToolJig*, a researcher can specify details about a tool's expected inputs and outputs, operating-system environment, and auxiliary files (e.g., scripts, configuration files). Researchers can also create workflows that integrate these tools. *ToolJig* checks the information provided by the user to ensure it is complete and valid. After successfully describing a tool or workflow, the researcher can download CWL files and use *ToolJig* to create 'input-object' files, which store input values for a particular invocation of a tool or workflow. In addition, *ToolJig* provides examples of how to execute the tool or workflow via a workflow engine.

## Using containers to manage software installation and configuration

First, we address software installation and configuration, which are essential steps to creating CWL tools and workflows. These seemingly simple steps are fraught with challenges. Although the software may be downloadable from a public website, installation instructions are sometimes vague, the process may involve many steps, and these steps may differ for each operating system. Such challenges led computational biologist Ian Holmes to quip, "You can download our code from the URL supplied. Good luck downloading the only postdoc who can get it to run" (**Holmes, 2013**). Package managers, such as *bioconda* and *bioconductor*, have helped to ameliorate these challenges, providing mechanisms for installing software dependencies and tracking versions. These package managers function in a way that is mostly agnostic to the user's operating system (**Grüning et al., 2018**; **Huber et al., 2015**). However, some software tools are not available via package managers, package managers may depend on operating-system components that cannot be installed using the package managers themselves, and package managers may not guarantee that older versions of software and their dependencies remain available (**Piccolo and Frampton, 2016**). Software *containers* have gained popularity among scientists in recent years because they help to overcome these limitations.

Software containers provide a mechanism to encapsulate specific versions of software and their dependencies in a fully configured, operating-system environment (**Boettiger, 2015**). Here, we focus on the *Docker* ecosystem, which is commonly used for building, managing, and sharing software containers (**Boettiger, 2015**). Other containerization systems are also available (https://coreos.com/rkt; **Gomes et al., 2018**; https://github.com/hpc/charliecloud, **Charliecloud collaborators, 2021**; **Priedhorsky and Randles, 2017**; https://podman.io); these systems are typically compatible with Docker. In academic-computing environments, such as university-run cluster computers, Singularity has gained popularity because containers can be executed without administrative privileges (**Kurtzer et al., 2017**).

Steps for configuring the operating-system environment and installing software within a container are documented in a 'Dockerfile'. Using such a file, researchers can build a container *image*, a layered set of operating-system components. Commonly, the base layer is a minimal implementation of a Linux distribution (e.g., Debian 10.3 or Ubuntu 18.04). Subsequent layers consist of software dependencies, configuration files, environment variables, etc. Once a container image has been created, it is portable and immutable. This is advantageous for computational reproducibility because one researcher can share an image with another researcher and know that its contents have remained static.

Docker Hub (https://hub.docker.com) is a common way to share container images. After building an image on a local computer, a researcher can 'push' the image to Docker Hub using a single command. Subsequently, others can 'pull' the image and reuse it on a different computer (again, using a single command). Tags can be attached to each image for versioning. Currently, Docker Hub is free to use when specific requirements have been met. The Docker engine also provides a 'registry' tool that enables individuals or organizations to host their own registry of container images. For example, Red Hat, Inc offers https://quay.io, which hosts the container images from the BioContainers project (**da Veiga Leprevost et al., 2017**).

A software *container* is an actively executing instance of a particular container image. Multiple containers of the same image can be executing simultaneously on the same computer (or different computers). Docker containers are always Linux-based; this is convenient for biology research because bioinformatics software is predominantly designed for Linux. But even though a container is Linux-based, it can be executed on non-Linux operating systems, such as Windows or Mac OS, via a container *engine*. Container engines use virtual machines to facilitate this interaction (*Piccolo and Frampton, 2016*).

In addition to packaging scientific software, container images can package analysis code. For example, upon analyzing a given dataset, a researcher may wish to share the code with the research community. Many researchers post analysis code on websites such as GitHub (https://github.com) or Open Science Framework (*Foster and Deardorff, 2017*). This practice can enable others to verify and reuse the code; it also benefits the original researcher who otherwise might lose track of analysis details (*Piccolo and Frampton, 2016*). But even when analysis code resides in the public domain, third-party researchers may experience difficulty executing it. Scripting languages like Python (RRID:SCR_008394) and R (R Project for Statistical Computing, RRID:SCR_001905) require interpreters. Analysis scripts may depend on specific versions of interpreters, but the third-party researcher may have a different version on their computer. In addition, most analyses rely on ancillary software packages. Such packages provide logic for parsing a certain type of file, performing statistical tests, creating graphics, etc. Versioning is critical: older (or newer) versions of software packages may be incompatible with the analysis code. Researchers can facilitate reproducibility by providing container images that include specific versions of any script interpreters or software packages that are necessary to execute an analysis.

*Binder* facilitates the containerization process for analysis code stored in GitHub repositories (*Project Jupyter et al., 2018*). To use Binder, a researcher creates a configuration file that indicates which software is needed in the container image. For Python and R analyses, these configuration files indicate packages that must be installed, as well as version information. In other cases, a Dockerfile can be used to configure the environment more flexibly. After the researcher places the configuration file (and analysis code) in a GitHub repository, other researchers can visit the Binder website and re-execute the analysis. Behind the scenes, Binder provisions a cloud-based computer and executes the code within a container. This solution is effective for relatively short-running analyses that require modest computational resources, that use small datasets, and that are ready to be released publicly. However, many analyses do not meet these criteria. Longer, more data-intensive analyses are a better fit for workflow engines.

## Basic elements of a CWL tool description

To describe execution of a command-line tool, a researcher creates a text-based file according to the CWL Command Line Tool Description specification (*Amstutz et al., 2016*). CWL files can be structured using either the YAML or JSON data-serialization formats (https://yaml.org; https://www.json.org). Here, we provide an overview of key components of CWL tool descriptions.

A CWL tool describes inputs that will be used when the command-line tool is executed. A data type (or schema) should be defined for each input, indicating whether it represents a string, number, Boolean value, file, directory, or array (a data structure with multiple values). In practice, these inputs are generally data files and configuration settings for the software. These definitions help users of the tool understand the nature of each input and make it easier for inputs to be validated. For example, if a command-line tool expects a particular input value to be an integer (e.g., number of threads), a workflow engine can verify that the user has specified an integer before executing the tool.

After inputs have been defined, a tool description must provide instructions for constructing a command from the inputs. These commands can be based on a sorted ordering of the inputs. Alternatively, the researcher can specify a template for the command, using placeholders for the inputs. Such templates can represent either a single command or multiple commands.

As a tool executes, it can generate three types of outputs that might be useful to a researcher: (1) standard output, (2) standard error, and (3) new files. *Standard output* often consists of informational messages printed to the console; but it may also contain data to be used as input for another tool. *Standard error* typically consists of errors, warnings, or diagnostic information printed to the console. Many command-line tools produce new data files that have resulted from execution of the tool. A

CWL tool description must indicate which of these outputs are expected so that a workflow engine can "collect" them after executing the tool.

## CWL tool examples for printing simple command-line messages

In the GitHub repository that accompanies this article, we have provided example tool descriptions, formatted in YAML according to the CWL specification (https://github.com/srp33/ToolJig/tree/master/examples, copy archived at swh:1:rev:ae8d3b358ccc44e45604125257c5361d20c26832, *Stephen, 2021*). The first series of examples is stored in the hello subdirectory.

The first example, 01_hello.cwl, requires two inputs: (1) a person's given name and (2) the person's surname. It uses the baseCommand field to construct a command based on ordinal positions specified for the inputs; the resulting command prints a greeting for that person. In this simple example, the only output is the greeting sent to standard output, which is redirected to a file called 01_output.txt. In the GitHub repository, hello_objects.yml is an example input-object file for this tool. It species a person's given name ('Fernanda') and surname ('Dantas'). The user could execute the tool via the *cwl* workflow engine using this command: cwltool 01_hello.cwl hello_objects.yml. The output would be, 'Hello, Fernanda Dantas'.

Suppose we wished to alter the greeting to include an exclamation point and to indicate the person's age. First, we would add an input for the person's age (an integer). Second, we would update the command that will be executed. However, CWL's baseCommand field provides limited flexibility for constructing commands. Instead, our example provides a command template using the ShellCommandRequirement and arguments fields and uses placeholders within the template for each input. As shown in 02_hello.cwl (and *Figure 1A*), we use the following template: echo Hello, $(inputs.given_name) $(inputs.surname)! You are $(inputs.age) years old.

We use a $ character and parentheses to indicate placeholders for input variables. We prefix each input variable with 'inputs' to indicate that they will be specified as inputs.

By default, the above two examples would be executed within the same operating-system environment as the workflow engine. Accordingly, these tools could only be executed on operating systems that support the echo command. Many commands, including echo, are only supported on particular operating systems—or their behaviors differ by operating system. So in 03_hello.cwl (and *Figure 1B*), we use a Docker image based on the 'buster' release (version 10.3) of the Debian Linux operating system. The DockerRequirement field is added (in this case, two lines of text). Before executing the tool, a researcher would install a container engine such as Docker Desktop (RRID:SCR_016445). Then, when executing the tool, the workflow engine would integrate itself with the container engine, which would identify any input files or directories and create container *volumes* so that the files or directories could be accessed from within the container. When a command-line tool executes within a Docker container, it becomes portable—it can be executed on any computer that supports the container and workflow engines.

## CWL tool examples for performing simple data analyses

The second series of examples is stored in the examples/bmi subdirectory on the GitHub site. The 01_bmi.cwl tool provides a simple example of a reproducible, quantitative analysis that could be performed using CWL. It accepts as input a tab-separated file containing names, weights, and heights of (fictional) individuals. A second input specifies the name of the column in the tab-separated file that contains weight information. The third input specifies the column name containing height information. The fourth input is the name of an output file that will be created. This example illustrates the use of an *auxiliary file*. Under the InitialWorkDirRequirement field, the contents of a Python script are stored. This script is used to calculate body mass index (BMI) values for each person in the input file and store those values in a *BMI* column in the output file. Below is the command template that we use.

```
python  calculate_bmi.py  "$(inputs.input_file.path)"  "$(inputs.weight_
column_name)" "$(inputs.height_column_name)" "$(inputs.output_file_name)"
```

The command template specifies the inputs as arguments to the Python script. When a file input is used, the workflow engine stores metadata about the file in an object with multiple attributes. Thus to reference the file's path within the command template, we append 'path' to the input name. As the workflow engine executes the tool, it stores the auxiliary Python script inside the container, invokes the script, and collects the output file that the script generates.

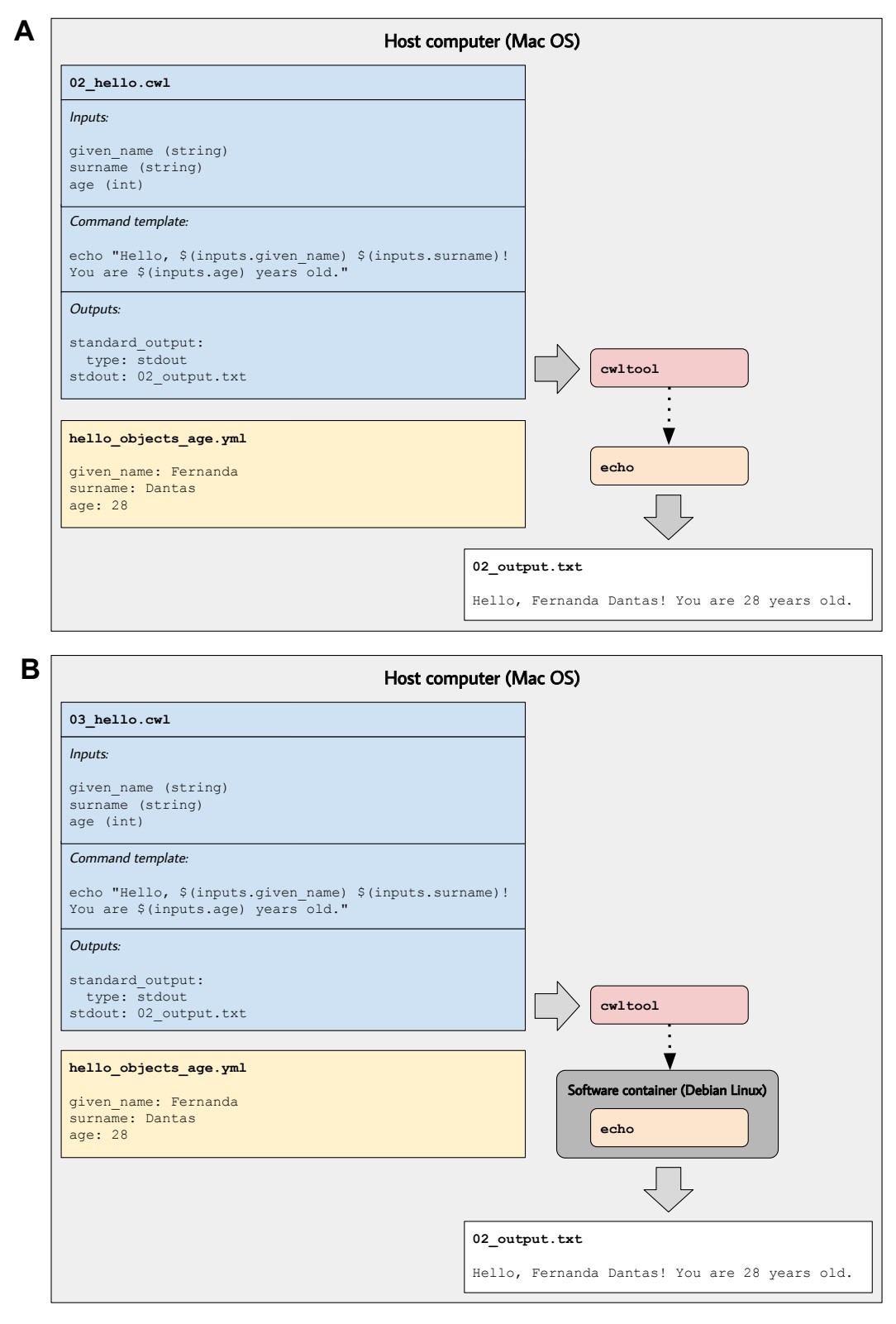

**Figure 1.** Illustration of tool descriptions for printing simple greetings. In the examples associated with this article, we provide tool descriptions that illustrate how to print custom greetings at the command line. These diagrams illustrate the 02_hello.cwl (**A**) and 03_hello.cwl (**B**) examples. In (**A**), the tool description indicates which inputs that must be specified, along with a template for executing the command; it also indicates that a message will be printed to standard output and that this message should be stored in a file called 02_output.txt. The hello_objects_age.yml input-object file stores

*Figure 1 continued on next page*

Figure 1 continued

values for a particular invocation of the tool. In (**A**), the cwltool workflow engine uses the host computer's operating system to execute the tool; thus, the echo command must be supported on that operating system. In (**B**), the tool description defines a software container environment; thus, cwltool executes the command within a container, which provides the echo command (packaged with the Debian Linux operating system).

If a researcher wished to ensure that others could reproduce the BMI calculations, they would need only to share the CWL file, the input-object file (01_bmi_objects.yml), and the data file (biometric_ data.tsv). However, many analyses use data stored in online repositories. In such cases, it is convenient for a CWL tool to pull data directly from an online repository. The 02_bmi.cwl tool description and *Figure 2* illustrate this approach. Similar to the previous example, it extracts names, weights, and heights from a tab-separated file and adds a BMI column. However, it pulls the file from a Web server (in this case, our GitHub repository). The command template is similar to the previous example. Again, we use a software container based on Debian Linux. But this time, we also use the NetworkAccess field to enable the container to connect to external computers. The tool emits messages to both standard output and standard error; these messages are stored in files called '02_output.txt' and '02_ error.txt', respectively.

## CWL tool examples for processing transcriptomic data

The third series of examples (examples/transcriptomic subdirectory) are wrappers around existing tools for processing transcriptomic data. In both cases, we use R packages from the Bioconductor suite (*Huber et al., 2015*). Although R and Bioconductor are designed to be compatible with all major operating systems, some packages require dependencies that are operating system-specific. Furthermore, many Bioconductor packages provide a large number of functions and options. Researchers can create CWL tool descriptions that install dependencies (within a container image) and support a narrower range of options. The researchers might then share this tool with other researchers, enabling them to apply the tool more easily to their own data. Alternatively, they might use the tool as a way to support reproducibility for their own analyses.

The single-channel array normalization (SCAN) algorithm normalizes data from gene-expression microarrays, correcting for background noise and oligonucleotide-binding biases (*Piccolo and Sun, 2012*). The SCAN method is implemented in the SCAN.UPC package in Bioconductor. It can download data directly from Gene Expression Omnibus (GEO) (*Barrett et al., 2011*). The scan_normalize.cwl example illustrates how this functionality could be incorporated into a CWL tool. The base container image in this example includes the core Bioconductor components; our Dockerfile extends this image by installing the SCAN.UPC package. In addition, our example uses an auxiliary file containing R code that invokes the SCAN function within this package to normalize a given GEO series. Upon executing, this tool produces a tab-separated output file containing normalized measurements for all biological samples in the series. The tool could be customized further, for example, to perform gene-level rather than probeset-level summarization or to perform a quality-control analysis.

Commonly, researchers seek to identify genes that are differentially expressed between two conditions. The DESeq2 package is popular for performing such analyses with RNA-sequencing data (*Love et al., 2014*). Our deseq2.cwl example illustrates how this process could be incorporated into a CWL tool. Similar to the previous example, this tool uses a container image with Bioconductor core components and installs the 'DESeq2' package. In addition, it installs the readr and dplyr packages (*Wickham, 2018a*; *Wickham, 2018b*), which we use to read and parse the data before identifying differentially expressed genes. The first two inputs are URLs to data files containing gene counts and phenotype information in tab-separated formats. The third input is a string representing a design formula; users of the tool can customize the differential-expression calculations based on the dependent variable of interest as well as any covariates. In the example input-object file (deseq2_objects.yml), we use data from an RNA-sequencing experiment that compared two inbred mouse strains commonly used for neuroscience research (*Bottomly et al., 2011*); the data had been aligned to a reference genome, and gene counts had been quantified previously.

## CWL workflow examples for performing mathematical calculations

The examples so far have illustrated how to execute command-line tools in isolation, whereas workflows execute multiple tools in defined sequences. CWL workflows must specify at least one input(s)

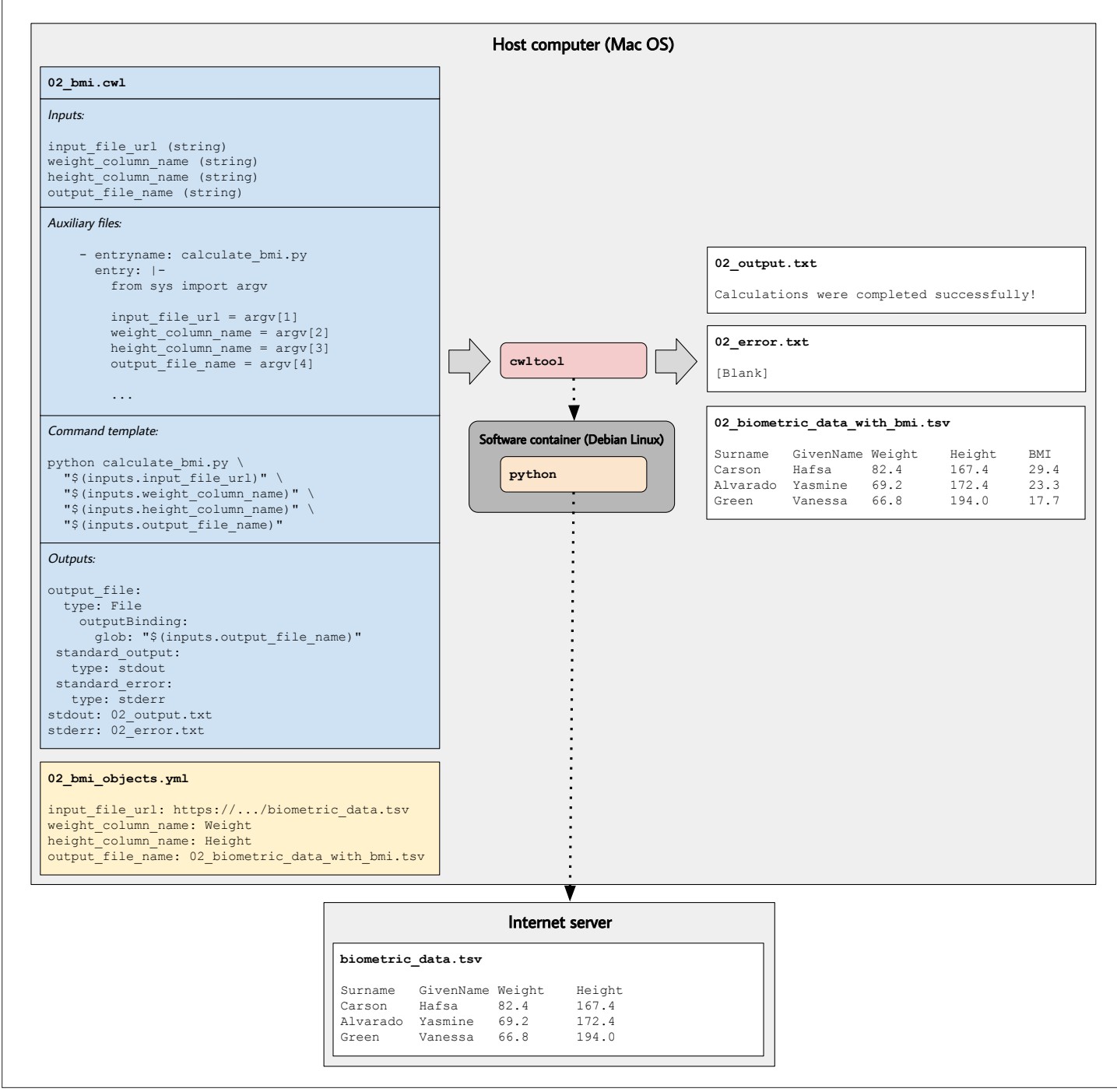

**Figure 2.** Illustration of tool descriptions for calculating individuals' body mass index (BMI). In the examples associated with this article, we provide tool descriptions that illustrate how to calculate BMI values based on individuals' weights and heights stored in a tab-separated value file. This diagram illustrates the 02_bmi.cwl example. The tool description indicates the expected inputs. In this case, the URL of a data file must be provided. That file must contain a column that stores weights (in kilograms) and a column that stores heights (in centimeters). In the input-objects file (02_bmi_objects.yml), the user specifies the names of these columns. The final input is the name of an output file that will be generated. This file will store the original data and a new column with the calculated BMI value for each individual. As the tool executes, Python (within a software container) downloads the input file, performs the calculations, generates the output file, and stores the standard output and standard error in text files.

and one output(s) for the entire workflow. In addition, the researcher must define steps that each consist of a tool with input(s) and output(s). The researcher indicates whether each step's input should be populated by an input for the entire workflow or by the output of a previously completed step. The workflow's output(s) consist of the output(s) of one or more of the steps. As with CWL tools, the researcher must create an input-object file that provides input values for a particular execution of the workflow. Upon executing the workflow, the workflow engine evaluates the sequence of steps that must be executed and connects inputs with outputs, as needed.

We provide three example workflows in the examples/workflows/math subdirectory of the GitHub repository. The add_sqrt_workflow.cwl example accepts two integers as inputs, sums them, calculates the square root of the sum, and then stores the square root of the sum in an output file. This example illustrates the basic process of using an output from one tool as input to another. The recursive_sqrt_workflow.cwl example reads a number from a file, calculates the square root of that number, calculates the square root of the resulting number, and saves the output to a file. This workflow demonstrates the ability to invoke the same tool recursively. The secondary_sqrt_workflow.cwl example calculates the square root of a number stored in a file and saves the result to an output file. It does the same for two secondary files. It then sums those values and writes the sum to a file. This example demonstrates using secondary files within a workflow. Secondary files are commonly used in genomics and simplify the process of working with groups of related files that are necessary to complete a particular task.

## CWL workflow example for identifying somatic variants in a cancer genome

The examples in the examples/workflows/somatic subdirectory demonstrate a process for calling somatic variants from Illumina sequencing reads. We use paired-end reads from tumor and normal cells for an individual from the Texas Cancer Research Biobank (*Becnel et al., 2016*). (Although these data are publicly available, they are subject to some data-use restrictions; *Becnel et al., 2016*.) To shorten execution times, we use a subset of the data: the first 10,000,000 reads from lane 2 of the sequencing run. Furthermore, our analysis is limited to essential steps for preparing the data and calling variants. Additional steps like annotation and filtering would improve sensitivity and specificity of the variant calls; accordingly, researchers should interpret our variant calls with caution.

In these examples, somatic-variant calling occurs in a series of steps (*Figure 3*):

1. *Download and index a human reference genome* (version hg38). We use the Linux wget and gunzip utilities to download and decompress a FASTA file from the UCSC genome repository (*Haeussler et al., 2019*). We also use bwa, samtools, and Picard Tools to index the FASTA file and create a sequence dictionary (*Li et al., 2009*; *Li and Durbin, 2009*; Picard, RRID:SCR_006525).
2. *Preprocess reference files containing known polymorphic sites in preparation for base-quality score recalibration* (BQSR). We download Variant Call Format files (*Danecek et al., 2011*) from the Genome Analysis Toolkit (GATK) resource bundle (*Depristo et al., 2011*) and use a custom Python script to adjust chromosome identifiers that may differ across reference genomes. We also use Picard Tools to sort the reference files.
3. *Download the raw sequencing reads* (FASTQ files). The files are stored in a publicly available, Open Science Framework repository (*Foster and Deardorff, 2017*).
4. *Trim adapter sequences and low-quality bases* using atropos (*Didion et al., 2017*).
5. *Align the trimmed reads to the reference genome* using bwa mem (*Li and Durbin, 2009*). A read-group string is also specified.
6. *Sort and index the resulting BAM files* (*Li et al., 2009*) using sambamba (*Tarasov, 2015*).
7. *Mark duplicate reads and re-index the resulting BAM files* using sambamba.
8. *Derive a BQSR table from each BAM file* using GATK.
9. *Apply the BQSR table to the BAM files* using GATK.
10. *Call somatic single-nucleotide variants and small insertions/deletions* using Mutect2 (*Benjamin et al., 2019*). This step produces a VCF file.
11. *Call somatic structural variants* using Delly (*Rausch, 2012*). This step produces a VCF file.

This workflow executes many of the same steps for a normal DNA sample and a tumor DNA sample. These steps are independent of each other. Accordingly, when multiple computing cores are available, the workflow engine may execute these steps in parallel.

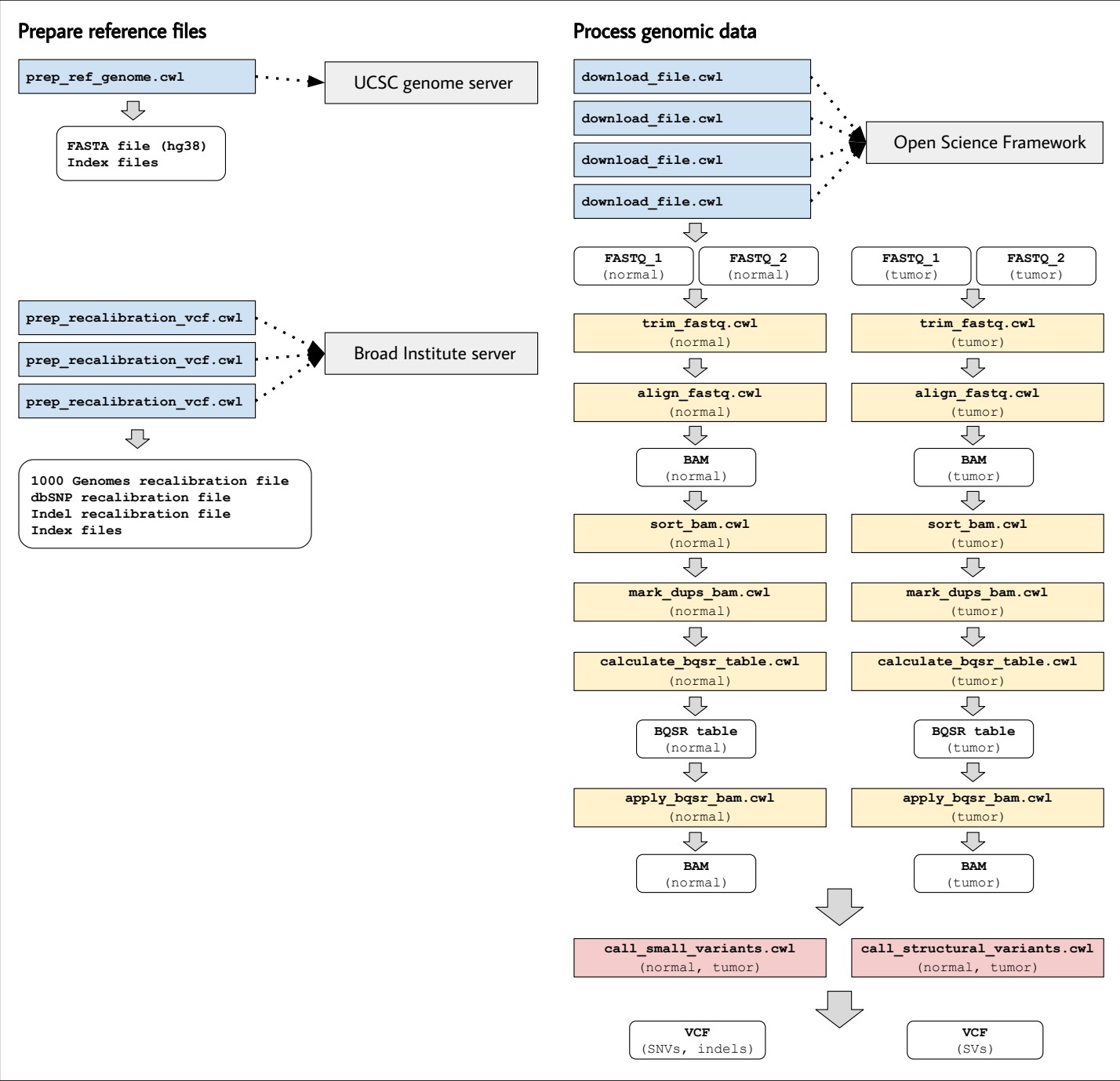

**Figure 3.** Illustration of tool descriptions for calling somatic variants from a cancer genome. In the examples associated with this article, we provide tool descriptions that illustrate how to call somatic variants from second-generation sequencing data for a cancer genome (compared against a normal genome from the same patient). This process requires execution of 11 distinct tools in a defined succession of steps (a workflow). Two tools (prep_ref_genome.cwl and prep_recalibration_vcf.cwl) prepare reference files associated with a given human reference genome. These tools download data files from public Internet servers and then create index files and standardize contig identifiers. The third tool (download_file.cwl) downloads FASTQ files from an Internet server. The remaining tools process the normal and tumor sequences separately before comparing the tumor genome against the normal genome to identify single-nucleotide variants, indels, and structural variants.

Most of the tools in this workflow use container images from the BioContainers project, which provides thousands of Docker images that encapsulate biology-related software (*da Veiga Leprevost et al., 2017*). In some cases, we used a BioContainers image that had been built for a specific version of a software package in the Bioconda project (*Grüning et al., 2018*). In cases where we used multiple

packages for a given task, we started with a base image from BioContainers and used Bioconda to install the packages. In the case of GATK, we used container images that had been created by the Broad Institute and stored on Docker Hub.

In most cases, we followed the recommendation that a single container image use only a single software package (*Gruening et al., 2019*). However, in some cases, we determined that it was more sensible to use multiple software packages in a single CWL tool. For example, when preparing index files for the reference genome, we use three separate software packages. In contrast, sometimes we used the same software in multiple CWL tools. For example, the BQSR steps are computationally intensive; thus, we separated them into distinct CWL tools so that computational resources can be allocated at a more granular level. In this sense, each CWL tool represents a practical unit for data processing, not necessarily a particular software package.

## Materials and methods
### Using ToolJig to create CWL tool descriptions
In manufacturing, a 'jig' is used by toolmakers to ensure that products are created in a repeatable, consistent pattern. Similarly, ToolJig provides a means to generate CWL tool descriptions, workflows, and input-object files in a repeatable, consistent manner. ToolJig is a Web application that uses the Vue.js framework (https://vuejs.org). Its functionality is divided into two pages: one for creating CWL tools and one for creating workflows. It is available from https://srp33.github.io/ToolJig. To create a tool description in ToolJig, users specify the following:

1. A unique identifier. This identifier is used in the name of the CWL file that is generated, as well as for tagging the Docker image.
2. A short label that describes the tool's purpose and function.
3. Optionally, a longer description that provides more detailed documentation about the tool.
4. Dockerfile contents. These instructions indicate the base container image that should be used and any additional commands necessary to build and configure a container image for the tool. A tutorial by Nüst, et al. provides recommendations on authoring Dockerfiles (*Nüst et al., 2020*).
5. Author information. Optionally, the tool author can specify their name and ORCID identifier (*Haak et al., 2012*). This information helps to ensure that authors are credited for their work.
6. Software license. The tool author can select from among seven popular licenses, thus indicating conditions under which others can use the CWL document. This license may or may not be identical to the license specified for the software itself.
7. Inputs. Users specify a name, type, and description for each input. Supported types are integer, string, File, and 'Output File'. The File type allows the user to indicate that an input file is expected and asks the user to specify the EDAM format of the file (*Ison et al., 2013*). Additionally, input files may be associated with secondary files. For instance, as our examples illustrate for somatic-variant calling, BAM files must be accompanied by index files. Rather than specify these as two separate inputs, we indicate that an index file is secondary to a BAM file. 'Output File' is a convenience type that is used when a tool author wants users to be able to specify the *name* of an output file that will be generated as the tool executes. Because this requires user input, ToolJig provides it as an input option. When a tool author specifies this type, ToolJig creates a *string* input for the file name, along with a corresponding output *file*, thus simplifying this process for the user.
8. Auxiliary files. The tool author can enter the name and contents of any auxiliary files that will be used. Commonly, these tools are programming scripts and can be written in any programming language that the Docker image supports.
9. Command template. The tool author specifies a template for executing the tool at the command line. Each input must be specified at least once in this template; ToolJig provides syntax suggestions to the user. ToolJig uses these command templates as an alternative to the baseCommand field. As our examples illustrate (*Figure 4*), command templates provide flexibility in the ways that commands are constructed and support the use of multiple commands. They provide the additional benefit that inputs do not have an inherent order and thus can be specified in the command template in any order (and multiple times, if desired).
10. Outputs. Aside from any 'Output Files' that may have been specified as inputs, the user may declare output files. For instance, the somatic/trim_fastq.cwl example specifies that trimmed FASTQ files should have the same names as the corresponding input FASTQ files. To indicate

**A**
```
echo "Hello, $(inputs.given_name) $(inputs.surname)! You are $(inputs.age) years old."
```

**B**
```
Rscript run_deseq2_analysis.R "$(inputs.read_counts_url)" "$(inputs.phenotypes_url)" \
    "$(inputs.design_formula)" "$(inputs.output_file_name)"
```

**C**
```
bwa mem -t $(inputs.threads) $(inputs.args) \
    -R "$(inputs.read_group_string)" \
    "$(inputs.fasta_file.path)" \
    "$(inputs.fastq_file_1.path)" \
    "$(inputs.fastq_file_2.path)" | samtools view -b > "$(inputs.output_file_name)"
```

**D**
```
sambamba sort -t $(inputs.threads) -o "$(inputs.output_file_name)"
"$(inputs.bam_file.path)"

sambamba index -t $(inputs.threads) "$(inputs.output_file_name)"
```

**E**
```
wget $(inputs.exclude_template_url)

delly call -x `basename "$(inputs.exclude_template_url)"` \
    -o output.bcf -g "$(inputs.fasta_file.path)" \
    "$(inputs.tumor_bam_file.path)" "$(inputs.normal_bam_file.path)"

bcftools view output.bcf > "$(inputs.output_file_name)"
```

**Figure 4.** Examples of command templates used in Common Workflow Language (CWL) tool descriptions. These examples illustrate diverse types of command templates for configuring execution of CWL tools. In each example, placeholders are used for inputs. When the tools are executed, the placeholders are replaced with input-object values. (**A**) A simple greeting is printed to standard output. (**B**) An R script (stored as an auxiliary file within the tool description) is executed; this script performs a differential-expression analysis using the DESeq2 package. (**C**) The *bwa* software aligns FASTQ files to a reference genome and pipes the output to *samtools*; the output is then converted to BAM format. This example illustrates a scenario in which two complementary software packages are used to perform a data-analysis task. Although these packages could be incorporated into distinct CWL tools, we combine them because read alignment and BAM conversion are typically performed jointly. (**D**) The *sambamba* software sorts and then indexes a BAM file. (**E**) The *Delly* software identifies structural variants in a cancer genome. *Delly* can be configured to exclude telomere and centromere regions as well as unplaced contigs. This example downloads an exclusion file, invokes *Delly*, and converts the output to VCF format. Examples (**D**) and (**E**) illustrate additional scenarios in which related tasks are executed as practical units.

this, the user specifies a CWL-based expression: $(inputs.fastq_file.basename). Additionally, if the user wishes to collect standard output or standard error, they may specify the names of files that will store these messages.

After a user has specified all required elements, ToolJig generates a YAML document that conforms to the CWL specification; the user may download this document. ToolJig also generates a form in which the user can indicate a value for each input. Subsequently, the user can download an input-object file in YAML format.

## Using ToolJig to create CWL workflows

When using ToolJig to create a workflow, researchers first enter metadata: a unique identifier, label, description, author information, and software license. Subsequently, the researcher uploads at least one tool description. The researcher then defines two or more workflow steps. For each step, they specify a unique name and the tool that will be used in that step. For each of the tool's inputs, the researcher indicates whether the input will be populated from the output of a preceding step or from

```
A   dockerPull: python:3.8.2-slim-buster

B   dockerImageId: prep_recalibration_vcf
    dockerFile: |-
        FROM quay.io/biocontainers/picard:2.22.3--0

C   dockerImageId: call_structural_variants
    dockerFile: |-
      FROM biocontainers/biocontainers:v1.0.0_cv4

      RUN conda install -c bioconda/label/cf201901 delly bcftools -y

D   dockerImageId: scan_normalize
    dockerFile: |-
        FROM bioconductor/bioconductor_docker:RELEASE_3_10

        RUN R -e 'BiocManager::install("SCAN.UPC")'
```

**Figure 5.** Examples of DockerRequirement specifications used in Common Workflow Language (CWL) tool descriptions. These examples illustrate diverse ways to configure CWL tools to be executed in software containers. In (**A**), a container image is pulled from Docker Hub; this image encapsulates a minimal ('slim') version of Debian Linux 10.3 ('buster') and includes the Python 3.9 interpreter. In (**B**), the contents of a Dockerfile are included within the CWL description. In this case, the Dockerfile is simple—it pulls an existing image from https://quay.io. This image is provided as part of the BioContainers project and includes the Picard Tools software. (**C**) uses a base image from BioContainers and the Bioconda package manager to install the *Delly* and *bcftools* software within the image. (**D**) uses a base image from Bioconductor and executes R code to install the SCAN.UPC package within the image.

a workflow input. The user may also indicate that any of the tool's outputs will become outputs for the overall workflow. As with tool descriptions, ToolJig validates the user's input and then generates a CWL document and input-object file that can be downloaded.

## Discussion

Progress in biology research is hindered when software tools are difficult to install, when inputs and outputs are inadequately or inconsistently specified, and when it is difficult to combine tools into workflows. The CWL specifications—in combination with package managers and software containers—are helping to alleviate these longstanding challenges. Moreover, CWL tool and workflow descriptions can facilitate reproducible research. Rather than simply providing analysis code alongside journal manuscripts, researchers can provide CWL documents. As illustrated in our examples, CWL documents provide instructions for executing analyses in software containers that encapsulate all relevant dependencies (*Figure 5*), along with ancillary scripts and instructions for accessing data files. People who read (or review) the manuscripts can then repeat the analyses, without needing to install any software other than a relevant workflow engine and container engine, even if their operating system or configuration differs from the authors'.

Multiple online repositories provide CWL documents, including the Dockstore tool registry (*O'Connor et al., 2017*) and GitHub. For example, a search on GitHub for CWL documents that use the FastQC software (*Brown et al., 2017*) resulted in 667 matches (August 30, 2021). Researchers can

reuse and adapt these documents. However, in cases where reuse is infeasible or extensive adaptations are necessary, scientific progress may be accelerated as researchers, including non-bioinformaticians, gain greater efficiency in creating CWL documents. ToolJig aims to facilitate this process, enabling researchers to build CWL tools interactively, without needing to gain a deep understanding of the CWL specification or YAML syntax.

One advantage of CWL is that it can be used with diverse workflow engines. Whether or not they support CWL, most workflow engines provide custom languages or programming interfaces for creating workflows. Relatively little support is available for migrating from these engine-specific solutions to CWL in an automated manner. However, when these engines support execution within Docker-compatible containers, researchers can migrate these tools manually using ToolJig (or other means). Providing better support in existing workflow engines for exporting to CWL will be a positive step toward ensuring that CWL truly becomes a common language for command-line tools and workflows.

The CWL specification provides considerable flexibility for describing command-line tools and workflows. Our goal was to support common use cases for biology research. For the sake of simplicity and to reduce barriers of entry for new creators of CWL documents, ToolJig does *not* support some optional features within the CWL specification. These include input directories, dependent and exclusive parameters, process requirements, hints, and output directories. The CWL specifications provide details about these features.

ToolJig has no dependencies other than a modern Web browser. Accordingly, it can be used from virtually any computer with no installation process. When ToolJig is updated, the user simply needs to refresh their browser. A tradeoff to this simplicity is that ToolJig does not provide a direct means of testing tools or workflows. However, the cwltest utility provides a command-line testing framework, enabling researchers to compare tool and workflow outputs with expected results. In particular, researchers implementing CWL in production systems would benefit from using such a utility for validation.

## Conclusions

CWL documents can formalize execution of command-line tools and workflows. We have summarized the key components of these documents and provided examples to illustrate key concepts. In addition, we have described ToolJig, a Web application that enables researchers to create CWL documents interactively. We hope these resources will benefit researchers from diverse training backgrounds to more easily create CWL documents and thus advance sharing of methods and computational reproducibility.

## Acknowledgements

We acknowledge the Texas Cancer Research Biobank and Baylor College of Medicine Human Genome Sequencing Center for providing cancer-genome data used in some of our examples.

## Additional information

### Funding

| Funder | Grant reference number | Author |
|---|---|---|
| National Institutes of Health | U54CA209978 | Stephen R Piccolo<br>Zachary E Ence<br>Jeffrey T Chang<br>Andrea Bild |

The funders had no role in study design, data collection and interpretation, or the decision to submit the work for publication.

### Author contributions

Stephen R Piccolo, Conceptualization, Funding acquisition, Software, Supervision, Visualization, Writing - original draft, Writing – review and editing; Zachary E Ence, Elizabeth C Anderson,

Conceptualization, Software, Writing – review and editing; Jeffrey T Chang, Conceptualization, Funding acquisition, Writing – review and editing; Andrea H Bild, Funding acquisition, Writing – review and editing

### Author ORCIDs
Stephen R Piccolo (iD) http://orcid.org/0000-0003-2001-5640

### Decision letter and Author response
Decision letter https://doi.org/10.7554/eLife.71069.sa1
Author response https://doi.org/10.7554/eLife.71069.sa2

## Additional files

### Supplementary files
• Transparent reporting form

### Data availability
We did not generate data for this manuscript. However, we did create software for this manuscript. The full code for the software are available on GitHub using a liberal, open-source license: https://github.com/srp33/ToolJig (copy archived at https://archive.softwareheritage.org/swh:1:rev:ae8d3b358ccc44e45604125257c5361d20c26832).

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
