## [Decision Letter]

**Acceptance summary:**

In this manuscript, Piccolo and collaborators provide a detailed overview of the Common Workflow Language (CWL) for beginner bioinformaticians, and perhaps more experienced workers that may not be up-to-date with the latest developments in reproducible research. They also provide a tool, ToolJig, to create CWL documents without needing to install any software nor to learn the specifications of the format. Written in the form of a tutorial, its major strengths are that explanations are very clear, and are accompanied by illustrative figures and examples in a Github repository. As science is currently undergoing a major reproducibility crisis, we think that it is crucial that detailed and accessible pieces such as this one are published to teach scientists to create fully reproducible code.

**Decision letter after peer review:**

Thank you for submitting your article "Simplifying the development of portable, scalable, and reproducible workflows" for consideration by *eLife*. Your article has been reviewed by 2 peer reviewers, including C Daniela Robles-Espinoza as Reviewing Editor and Reviewer #1, and the evaluation has been overseen by Aleksandra Walczak as the Senior Editor.

This is an accessible tutorial that illustrates the basics of Common Workflow Language (CWL), intended both for students beginning their scientific formation and for more experienced bioinformaticians that have not adopted this community standard yet. The adoption of CWL has the potential to help alleviate the reproducibility crisis ongoing in scientific publishing. The reviewers have agreed that the figures are very clear, and the examples available on Github are helpful and they go from simple tasks to illustrating a more complex biological analysis. A webtool is also provided for easily creating CWL documents. There are a few revisions that the reviewers have recommended to make the work clearer before publication:

1. There is some confusion regarding the usage of containers in regards to the location of the workflow manager (production vs publication workflows). A container with all required analysis software, CWL document and a workflow manager seems well suited for distribution with a publication for reproducible calculations. However for production needs, a modular design with various containers and a workflow manager outside of the containers seem a better choice. It's hard to distinguish these two usages in the manuscript. Can a few lines be devoted to making these clarifications, please?

2. A reviewer notes, "I think it would be useful to beginners to see more information on the benefits of using CWL over some of the alternatives (Nextflow, Snakemake). There are quite a few ways of handling bioinformatic pipelines and in my experience I been overwhelmed by the decision of having to choose one over the rest.". Can a few lines be devoted to discussing the benefits/weaknesses of these other alternatives?

3. There is no mention of the benefits of error handling and restarts when using a workflow manager. For production environments this is a key benefit, so could this be incorporated please?

4. A reviewer mentions. "The manuscript does a good exposition of Docker and containers. However, I didn't find much mention of Docker Hub. Being that Docker Hub was a key element in the prominence of containers I think there should be a bit more details about it on the paper.". Could some lines about Docker Hub be incorporated please?

5. About figures 1 and 2 – should the name of the program in the yellow square be changed to match the .yml found in the GitHUb repository?

6. Would it be possible to add a button to ToolJig that would populate the fields, as a pre-filled example? It may be easier for beginners to illustrate how the information should be input. Perhaps following the example of one of the ones that are already in the GitHub repository so it's more easily comparable.

7. Could you please specify If the YAML document created by a user is saved by ToolJig, or is it deleted when the user closes the webpage?

---

## [Author Response]

1. There is some confusion regarding the usage of containers in regards to the location of the workflow manager (production vs publication workflows). A container with all required analysis software, CWL document and a workflow manager seems well suited for distribution with a publication for reproducible calculations. However for production needs, a modular design with various containers and a workflow manager outside of the containers seem a better choice. It's hard to distinguish these two usages in the manuscript. Can a few lines be devoted to making these clarifications, please?

Yes, thank you. We have reworked and expanded the second paragraph of the Introduction to address these points. We have pasted this paragraph below for your convenience. (We use the term "orchestration systems" rather than "production systems" because the former appears to be used more commonly in the literature.)

"In scientific research, the use of workflows can be classified into two main categories. One category includes orchestration systems, in which workflow engines repeatedly process data of a given type. […] Workflows aid in this process because they provide a way to encapsulate the logic for executing code in a fully specified computational environment with all necessary software dependencies installed. For many studies, a relatively simple workflow engine like cwltool can be used to re-execute the analyses."

2. A reviewer notes, "I think it would be useful to beginners to see more information on the benefits of using CWL over some of the alternatives (Nextflow, Snakemake). There are quite a few ways of handling bioinformatic pipelines and in my experience I been overwhelmed by the decision of having to choose one over the rest.". Can a few lines be devoted to discussing the benefits/weaknesses of these other alternatives?

Yes, thank you. We have added text to the Introduction section on this topic, as shown below:

"Different workflow engines use different methodologies and vocabularies for defining workflows and for interfacing with software within those workflows (Leipzig, 2017). […] Indeed, CWL documents are already recognized by many workflow engines, including Snakemake (Köster and Rahmann, 2012), cwltool, Toil (Vivian et al., 2017), Apache Airflow (Kotliar et al., 2019), Tibanna (Lee et al., 2019), and Arvados (Amstutz, 2015)."

3. There is no mention of the benefits of error handling and restarts when using a workflow manager. For production environments this is a key benefit, so could this be incorporated please?

Yes, thank you. We have added text to the Introduction section on this topic, as shown below:

"Scientists might also take advantage of error handling provided by some workflow engines. For example, if a power outage occurred while a workflow was being executed, it would be possible to restart the workflow at the point of failure rather than needing to rerun the entire workflow."

4. A reviewer mentions. "The manuscript does a good exposition of Docker and containers. However, I didn't find much mention of Docker Hub. Being that Docker Hub was a key element in the prominence of containers I think there should be a bit more details about it on the paper.". Could some lines about Docker Hub be incorporated please?

Yes, thank you. We have added a paragraph on this topic to the "Using containers to manage software installation and configuration" subsection, as shown below:

"Docker Hub (https://hub.docker.com) is a common way to share container images. After building an image on a local computer, a researcher can “push” the image to Docker Hub using a single command. Subsequently, others can “pull” the image and reuse it on a different computer (again, using a single command). Tags can be attached to each image for versioning. Currently, Docker Hub is free to use when specific requirements have been met. The Docker engine also provides a “registry” tool that enables individuals or organizations to host their own registry of container images. For example, Red Hat, Inc offers https://quay.io, which hosts the container images from the Biocontainers project(da Veiga Leprevost et al., 2017)."

5. About figures 1 and 2 – should the name of the program in the yellow square be changed to match the.yml found in the GitHUb repository?

Thank you for noticing this. We have modified Figures 1 and 2 to be consistent with what we have in the GitHub repository.

6. Would it be possible to add a button to ToolJig that would populate the fields, as a pre-filled example? It may be easier for beginners to illustrate how the information should be input. Perhaps following the example of one of the ones that are already in the GitHub repository so it's more easily comparable.

Thank you for the suggestion. We want to make sure the tool is as useful and user-friendly as possible. We have added "Show/hide example" links to ToolJig. These links enable users to see examples of what to specify.

7. Could you please specify If the YAML document created by a user is saved by ToolJig, or is it deleted when the user closes the webpage?

The YAML document created by a user is *not* saved by ToolJig. It will be deleted when the user closes the webpage. However, if they download the file, they can upload it back into ToolJig. We have added a comment to the app to clarify this for users.